# Exploring Prokaryotic and Eukaryotic Microbiomes Helps in Detecting Tick-Borne Infectious Agents in the Blood of Camels

**DOI:** 10.3390/pathogens10030351

**Published:** 2021-03-16

**Authors:** Wessam Mohamed Ahmed Mohamed, Alsagher O. Ali, Hassan Y. A. H. Mahmoud, Mosaab A. Omar, Elisha Chatanga, Bashir Salim, Doaa Naguib, Jason L. Anders, Nariaki Nonaka, Mohamed Abdallah Mohamed Moustafa, Ryo Nakao

**Affiliations:** 1Laboratory of Parasitology, Faculty of Veterinary Medicine, Graduate School of Infectious Diseases, Hokkaido University, Sapporo, Hokkaido 060-0818, Japan; wessam@czc.hokudai.ac.jp (W.M.A.M.); chatanga@vetmed.hokudai.ac.jp (E.C.); abassdoaa@yahoo.com (D.N.); nnonaka@vetmed.hokudai.ac.jp (N.N.); 2Division of Infectious Diseases, Animal Medicine Department, Faculty of Veterinary Medicine, South Valley University, Qena 83523, Egypt; alsagher.ali@vet.svu.edu.eg (A.O.A.); mhassan@vet.svu.edu.eg (H.Y.A.H.M.); 3Department of Parasitology, Faculty of Veterinary Medicine, South Valley University, Qena 83523, Egypt; dr_mosab2081@yahoo.com; 4Department of Veterinary Medicine, College of Agriculture and Veterinary Medicine, Qassim University, Buraidah 51452, Saudi Arabia; 5Department of Veterinary Medicine, Lilongwe University of Agriculture and Natural Resources, Lilongwe P.O. BOX 219, Malawi; 6Department of Parasitology, Faculty of Veterinary Medicine, University of Khartoum, Khartoum North 11111, Sudan; bashirsalim@gmail.com; 7Department of Hygiene and Zoonoses, Mansoura University, Mansoura 35516, Egypt; 8Graduate School of Environmental Science, Hokkaido University, Sapporo, Hokkaido 060-0810, Japan; janderslee@ees.hokudai.ac.jp; 9Division of Wildlife Medicine, Animal Medicine Department, Faculty of Veterinary Medicine, South Valley University, Qena 83523, Egypt

**Keywords:** *Candidatus* Anaplasma camelii, eukaryotes, microbiome, *Theileria*, Trypanosoma evansi

## Abstract

Dromedary camels (*Camelus dromedarius*) are widely distributed in Africa, the Middle East and northern India. In this study, we aimed to detect tick-borne pathogens through investigating prokaryotic and eukaryotic microorganisms in camel blood based on a metagenomic approach and then to characterize potentially pathogenic organisms using traditional molecular techniques. We showed that the bacteria circulating in the blood of camels is dominated by Proteobacteria, Bacteroidetes, Firmicutes and Actinobacteria. At the genus level, *Sediminibacterium*, *Hydrotalea*, *Bradyrhizobium* and *Anaplasma* were the most abundant taxa. Eukaryotic profile was dominated by Fungi, Charophyta and Apicomplexa. At the genus level, *Theileria* was detected in 10 out of 18 samples, while *Sarcocystis*, *Hoplorhynchus* and *Stylocephalus* were detected in one sample each. Our metagenomic approach was successful in the detection of several pathogens or potential pathogens including *Anaplasma* sp., *Theileria ovis*, *Th. separata*, *Th. annulate*, *Th. mutans*-like and uncharacterized *Theileria* sp. For further characterization, we provided the partial sequences of citrate synthase (*gltA*) and heat-shock protein (*groEL*) genes of *Candidatus* Anaplasma camelii. We also detected *Trypanosoma evansi* type A using polymerase chain reaction (PCR) targeting the internal transcribed spacer 1 (ITS1) region. This combined metagenomic and traditional approach will contribute to a better understanding of the epidemiology of pathogens including tick-borne bacteria and protozoa in animals.

## 1. Introduction

Camels are divided into three species, the Arabian or dromedary camel (*Camelus dromedarius*), the Bactrian camel (*Camelus bactrianus*) and the wild Bactrian camel (*Camelus ferus*) [1,2]. Arabian camels are commonly used for meat and milk production, transportation and racing [3]. There are many species of ticks that can infest camels including *Hyalomma anatolicum*, *Hy. excavatum*, *Hy. scupense*, *Hy. dromedarii*, *Hy. impeltatum*, *Hy. marginatum*, *Hy. rufipes*, *Hy. truncatum*, *Ornithodoros savignyi*, *Rhipicephalus praetextatus* and *Rh. turanicus* [4]. Such ticks can act as vectors for pathogens harbored by camels such as *Rickettsia*, *Anaplasma*, *Ehrlichia, Coxiella*, *Babesia*, *Hepatozoon* and *Theileria* [3,5,6,7,8,9], which represent a potential risk to other animals and humans. Despite the importance of camels in the livelihood of people, scarce information is available on their role in maintaining tick-borne pathogens (TBPs) as compared to other animals [10].

The metagenomic approach utilizing high-throughput sequencing techniques has contributed largely to the description of pathogenic and nonpathogenic microbes in many animal species. It is not limited to the simultaneous detection of known pathogens but also detection of novel potential pathogens [11]. The most common genetic markers targeted for metagenomic investigations are 16S rDNA and 18S rDNA for prokaryotes and eukaryotes, respectively [12]. However, polymerase chain reaction (PCR) using universal primers primarily amplifies the 18S rDNA of host animals rather than those of animal-associated eukaryotes, thereby limiting the use of metagenomics for such studies [13]. A PCR using non-metazoan universal primers (UNonMet-PCR), which was originally developed for the selective amplification of oyster protists [14], has been used to reduce the amplicons from metazoan 18S rDNA with high ability to amplify the V4 region of 18S rDNA from other eukaryotes within animal hosts [11]. Although this PCR can theoretically be used for many types of samples, it has not yet been applied for the detection of eukaryotes within the mammalian blood.

The study of blood parasites in mammals by conventional PCR or quantitative PCR can only detect genetically identified species [15]. However, next-generation sequencing (NGS)-based techniques can provide a thorough characterization of the eukaryome [16,17]. Furthermore, although it is thought that healthy mammalian blood should be sterile [18], some studies have provided evidence that there is a blood associated microbiome [19,20,21].

The present study aims to describe both the prokaryotic and eukaryotic profiles in the blood of dromedary camels in Egypt. The results were further verified by conventional genetic assays amplifying genus or species-specific genes. Our data indicate the usefulness of metagenomic approach for the detection of pathogens, but at the same time, suggest the necessity to refine current methods to capture a full picture of the blood microbiome in camels. 

## 2. Results

### 2.1. Bacterial Profile

The NGS targeting 16S rDNA resulted in a total number of 920,376 raw paired-end reads. DADA2 quality control analysis resulted in 403,680 high-quality paired-end reads with an average of 21,246 per sample with a maximum of 41,514 and minimum of 9,723. Sequences were divided into 3,772 amplicon sequence variants (ASVs). The possible contaminants in the negative control composed of two main phyla, the Proteobacteria and Firmicutes, which contain nine genera: *Pseudomonas*, *Escherichia*-*Shigella*, *Bacillus*, *Serratia*, *Novosphingobium*, *Curvibacter*, *Halomonas*, *Candidatus* Lariskella and *Coxiella* (Appendix A). After removing these contaminants from each sample, we finally obtained 333,054 reads divided into 2372 features. The mean frequency per sample was 18,503 and the median average was 16,856 reads. The most abundant phyla were Proteobacteria (mean = 46.7%), Bacteroidetes (mean = 24.1%), Firmicutes (mean = 16.9%) and Actinobacteria (mean = 8.2%). At the genus level, *Sediminibacterium* (mean = 13.8%), *Hydrotalea* (mean = 8.6%), *Bradyrhizobium* (mean = 7.9%), *Anaplasma* (mean = 6.4%) and *Ralstonia* (mean = 3.4%) were the highest abundant (Figure 1 and Appendix A). The genera *Sediminibacterium, Hydrotalea* and *Bradyrhizobium* were detected in all samples, while *Anaplasma* was detected in 11 samples (Table 1). Other known tick-borne bacteria including *Borrelia*, *Rickettsia* and *Ehrlichia* were not detected in our samples.

### 2.2. Eukaryotic Profile

A total number of 1,643,804 raw paired-end reads of 18S rDNA were obtained. A total of 728,071 high-quality paired-end reads remained after quality filtering using the DADA2 quality control analysis and classified into 353 unique features with a mean frequency per sample of 38,319 with a maximum of 52,151 and minimum of 10,668. The possible contaminants in the negative control composed mainly of fungi and Ochrophyta (Appendix A). In addition, the sequences of the most abundant taxon Metazoa (mean = 72.9%), which were derived from host DNA, were considered as contaminants. By filtering out the contaminants from each sample, a total of 189,914 reads divided into 275 features were obtained. The mean frequency per sample was 10,550 and the median average was 8933 reads. The most abundant taxa were Fungi (mean = 52.1%), Charophyta (mean = 38.8%), Apicomplexa (mean = 5.5%) and Cercozoa (mean = 1.3%). The phylum Apicomplexa was dominated by the genus *Theileria*, which was detected in 10 samples (Table 1). The genera *Sarcocystis*, *Hoplorhynchus* and *Stylocephalus* were detected in one sample each (Figure 2 and Appendix A). Other known tick-borne bacterial genera including *Babesia* and *Hepatozoon* were not detected in our samples.

### 2.3. Conventional Polymerase Chain Reaction (PCR) and Phylogenetic Relationships

We found that 29% (29/100) of the camels were positive by PCR targeting 16S rDNA of Anaplasmataceae (EHR-PCR). Partial gene sequences of the 16S rDNA from the 29 positive individuals were 99.7% (two samples) and 100% (27 samples) identical to Candidatus Anaplasma camelii (MT510533) from a dromedary camel in Laisamis, Kenya. The maximum likelihood (ML) phylogenetic tree based on the 16S rDNA sequences showed that Anaplasma sp. detected from this study clustered with the previously published Candidatus A. camelii and Anaplasma platys (Figure 3).

The partial sequences of the citrate synthase (gltA) gene obtained from 11 samples were 78.7% similar to A. platys (MH716422) from a tick in Shaanxi Province, China and and 77.4% to A. platys strain S3 (CP046391) from a dog in Saint Kitts and Nevis. The ML phylogenetic tree based on the gltA gene sequences showed the partial sequences from Anaplasma-positive samples formed an independent cluster separated from the other Anaplasma spp. (Figure 4). 

The partial sequences of the heat-shock protein (groEL) gene obtained from 11 Anaplasma-positive samples showed 99–100% identity to each other. These sequences showed 88.3% and 86.3% identity to A. platys (MH716435) found from a Rhipicephalus microplus tick from China and (CP046391) and a dog in Saint Kitts and Nevis, respectively. The obtained sequences were all clustered together in a separate clade from other Anaplasma spp. in the ML phylogenetic analysis (Figure 5).

PCR targeting 18S rDNA of piroplasma species (RLB-PCR) yielded amplicons in 39% (39/100) of the camels. However, the sequencing analysis revealed that the amplified RLB-PCR products did not belong to piroplasma species and were classified as Cercomonas species.

PCR targeting internal transcribed spacer 1 (ITS1) region of Trypanosoma species (ITS1-PCR) was positive for 19% (19/100) of the camels. All ITS1-PCR positive samples were also positive by PCR using the primers ILO7957 and ILO8091 (ILO-PCR) targeting the RoTat1.2 VSG region but negative by EVAB-PCR targeting a Tr. evansi type B-specific minicircle sequence, indicating that all Trypanosoma-positive samples were Tr. evansi type A.

### 2.4. Phylogenetic Analysis of the Detected Theileria Species

Conventional PCR targeting the 18S rDNA of piroplasma species did not amplify *Theileria* in our samples. We implemented a phylogenetic analysis of the genus *Theileria* using the obtained 18S rDNA sequences from NGS. Breifly, features belonging to the genus *Theileria* were exported and searched in BLAST to identify the species. The assigned taxonomic identities of these features by silva-132-99-nb classifier in QIIME2 were represented as follows: *Th. ovis* (*n* = 6), Th. separata (*n* = 2), *Theileria* sp. strain MSD (*n* = 2), *Th. annulate* (*n* = 1) and Theileria sp. (*n* = 1). Phylogenetic analysis showed that the detected *Theileria* spp. formed four clades clustered separately with Th. ovis, Th. separata, Th. annulate and *Th. mutans.* In addition, one *Theileria* sequence (LC592660) did not form a clade with *Theileria* sequences available in GenBank (Figure 6).

### 2.5. Comparison of Next-Generation Sequencing (NGS)-Based and Conventional PCR Results

Our results showed that the detection rates of *Anaplasma* and *Theileria* were higher using the NGS-based approach than the conventional PCR method. *Anaplasma* and *Theileria* were detected in 11 and 10 samples, respectively out of 18 samples using NGS. Conventional PCR was able to detect *Anaplasma* in 5 out of the 11 samples from which *Anaplasma* were detected by NGS-based approach. No *Theileria* spp. were detected by conventional PCR. However, conventional PCR could detect *Tr. evansi* in the examined camel blood samples that were negative for this parasite by NGS-based approach (Table 2). 

## 3. Discussion

Sudan and Ethiopia are the main source of camels to Egypt, with more than 750,000 camels imported between 2012 and 2015 [22]. This legal trade has contributed to the introduction of many infectious diseases such as Rift Valley fever (RVF) [23], Middle East respiratory syndrome coronavirus (MERS-CoV) infection [24] and theileriosis [10]. To the best of our knowledge, this study is the first to describe both the prokaryotic and eukaryotic profiles in the blood of camels. Our strategy was to detect potential pathogens by NGS-based approach in the randomly selected camel samples and to confirm the results by testing all camel samples using target-specific conventional PCR assays. 

We found that the blood microbiome of camels tested in this study is dominated by Proteobacteria which is different from the dominant bacterial phyla in the gut microbiome of flightless mammals (Bacteroidetes) [25,26]. This can be attributed to the high abundance of features belonging to the family Anaplasmataceae that represented the 4th dominating Proteobacteria in our samples after Pseudomonadaceae, Burkholderiaceae and Xanthobacteraceae (Appendix A). There are several potential sources for blood microbes in mammals. For example, it was found that bacteria can be translocated from the gastrointestinal tract to the blood through the intestinal epithelium [27]. However, it was demonstrated that the microbiome of human blood in healthy individuals is more similar to those of skin and oral cavity as opposed to the gut microbiome [28]. The translocation of bacteria from the skin and oral cavity to the blood likely occurs through open wounds. In addition, it is also possible that ticks introduce nonpathogenic microbes into the blood stream of camels as previously described in mice [29]. In addition, many other insects including mosquitoes, lice, fleas and true bugs can introduce several microorganisms [30]. However, there is still a possibility that environmental bacteria might have been introduced during sampling and laboratory procedures. Therefore, more experimental investigations are required to confirm the presence of these bacteria in camel blood by employing culture-based and target specific PCR-based methods.

Despite the detection of several eukaryotes using our NGS-based approach, we could not succeed in the complete elimination of camel-derived sequences. This caused the dominance of Metazoa (mean abundance of 72.9% before removing contaminants) in our samples, suggesting that increasing the depth of sequencing can obtain more eukaryotic members if they exist [13]. However, we were able to identify five different *Theileria* spp. using this NGS approach for the first time in camels. Moreover, it facilitated the detection of coinfections between several *Theileria* species (Table 1). The results of our study suggest that imported camels from Sudan could introduce several *Theileria* spp. to domestic animals in Egypt through the spillover effect. Recent studies reported *Th. ovis*, *Th. separata*, *Th. annulate* and *Th. lestoquardi* from domestic animals in Sudan [31,32]. Other studies reported that *Th. annulate, Th. lestoquardi*, *Th. equi*, *Th. ovis* and *Th. velifera* were detected in *Hy. anatolicum*, *Hy. Impeltatum* and *Amblyomma lepidum* [33]. These tick species can infest camels, cattle and sheep [4,34,35], suggesting that a possible transmission of these *Theileria* spp. may occur among the different livestock through tick vectors in Sudan. Moreover, several undescribed *Theileria* spp. were detected in camels [10] as well as *Th. annulata* from cattle in Egypt in 2014 [36]. Interestingly, one of the two detected undescribed *Theileria* spp. is closely related to *Th. mutans* while the other is separated from other *Theileria* spp. on our phylogenetic analysis (Figure 6). Although these *Theileria* spp. have not been reported before in Egypt, both were detected in camels from Ethiopia [37]. In this study, all *Theileria* spp. were detected from two or more camel samples except for one *Theileria* sp. and *Th. annulata* that were detected from only one sample each (CL-BL-1 and CL-BL-19, respectively). The number of *Theileria* reads obtained for these samples was 65 and 314, suggesting that they were not false positives. More investigations are required to explore the prevalence of the newly detected *Theileria* species in camels and ticks.

Although previous studies reported the detection of *Th. annulata*, *Th. equi* and *Th. ovis* in camel blood using conventional PCRs [9,38,39], our RLB-PCR assay did not work for the samples that were *Theileria*-positive by the NGS-based approach. One possibility is the presence of other eukaryotes that can be amplified by the same primer set. In fact, most of the sequences from RLB-PCR products were classified to the genus *Cercomonas* by Sanger sequencing method. To overcome this cross-reactivity issue, RLB-PCR products can be subjected to NGS. This strategy was applied to detect several *Theileria* spp. in African buffalo from South Africa [40] and piroplasm populations in wildlife and cattle in Zambia [41].

*Tr. evansi* type A was one of the most common parasites detected in camel blood in our study. Our NGS-based approach was not successful in detecting *Trypanosoma* spp. in the examined samples that were positive by conventional PCR. This is possibly due to a sequence mismatch between UNonMet-PCR primers and *Tr. evansi* 18S rDNA. We found that the forward primer (18S-EUK581-F) has a single nucleotide mismatch with the 18S rDNA sequences of *Tr. evansi* in GenBank. In addition, the failure of detecting *Tr. evansi* by NGS might be due to low parasitic load in the samples. Moreover, the high abundance of host DNA fragments can also contribute to hindering the amplification of *Tr. evansi*. Therefore, further investigations and experimental studies are required to improve this approach.

The prevalence of *Tr. evansi* infections detected in this study (19%) differed from what was previously published in Egypt [39,42,43], which reported extremely different prevalence in camels ranging from 4.7% [42] to 71% [43]. Although both studies used molecular techniques to detect this parasite from the blood of camels, the large difference in prevalence may be due to variation in geographic location of sample collection sites. 

There are several studies reporting *Candidatus* A. camelii in camels from Saudi Arabia [44], Kenya [45], Nigeria [7], Morocco [46] and Iran [47]. Our study showed that 29% of the camels imported from Sudan were infected with *Candidatus* A. camelii. In fact, this recently described bacterium has also been reported from cattle (*Bos taurus*) and deer (*Rusa timorensis*) in Malaysia [48], suggesting that it could spillover to a wide range of animals in Egypt. In a previous study, antibodies against *Anaplasma* were detected in camel blood in Egypt, however, species identification was not attempted [49]. The high sequence similarity of the 16S rDNA of *Candidatus* A. camelii and *A. platys* makes it difficult to describe the evolutionary relationship and host range of both species. This study provided partial sequences of *gltA* and *groEL* genes for the first time from *Candidatus* A. camelii. The sequences obtained were 78.7% and 88.3% identical to the *gltA* and *groEL* partial gene sequences of *A. platys*, respectively, suggesting that both genes are useful markers to differentiate between the two closely related *Anaplasma* species. Although the prevalence of *Candidatus* A. camelii was relatively high, this study could not confirm whether this bacterium can pose a potential veterinary or zoonotic risk.

Metagenomics of the host-associated eukaryotic populations has been challenging primarily because most universal primers amplify the 18S rDNA of host animals rather than those of animal-associated eukaryotes [13]. In the present study, we applied UNonMet-PCR to reduce the amplicons from metazoan, that is, camel. Other promising methods, which can be applied through exploring the eukaryotic microorganisms in mammalian hosts, include blocking primers with a C3 spacer addition at the 3′ end. This blocking method was also successfully used to bind the host DNA, resulting in blocking its amplification by inhibiting the elongation of the primer [50]. In addition, peptide-nucleic acid (PNA) blockers have been used to inhibit the amplification of 18S rDNA of non-target organisms including mosquitoes [51]. PNA blockers were also successful in increasing the sequence coverage of coral microbial communities by reducing host contamination [52]. These methods may facilitate metagenomic studies of host-associated eukaryotes within mammalian blood.

## 4. Conclusions

This study implemented a metagenomic investigation targeting the blood circulating prokaryotes and eukaryotes in camels, which could be used in other species of mammals. We revealed that the camels imported from Sudan were infected with a total of seven pathogens or potential pathogens, including *Tr. evansi* type A, *Candidatus* A. camelii, *Th. ovis*, *Th. separata*, *Th. annulata*, *Th. mutans* and undescribed *Theileria* species. In addition, we provided *gltA* and *groEL* partial gene sequences of *Candidatus* A. camelii that can be used to differentiate this pathogen from *A. platys*. Our study results support the hybrid approach to characterize the microbe of interest in mammals using NGS and conventional PCR combined with Sanger sequencing as was found in humans [53]. This will increase our understanding of the risks posed by importing animals and the accompanying tick-borne pathogens.

## 5. Materials and Methods

### 5.1. Camel Blood Samples and DNA Extraction

Blood samples were collected from the jugular vein of 100 camels from Abu Simbel near the Egypt–Sudan border. All camels were adult males imported from Sudan and kept in quarantine during the sampling process. The blood samples were collected on Na-EDTA tubes and DNA was extracted using innuPREP Blood DNA Mini Kit (Analytik Jena AG, Jena, Germany) following the manufacturer’s recommendations and stored at −20 °C until analyzed.

### 5.2. 16S rDNA and 18S rDNA Amplification and Illumina MiSeq Sequencing

A total of 18 camel blood samples were used for investigation by a metagenomic approach. Since all camels showed no clinical signs, we selected these samples randomly. We performed PCR targeting the V3-V4 region of the 16S rDNA of prokaryotes using Illumina barcoded primers (San Diego, CA, USA) Illumina_16S_341F and Illumina_16S_805R as recommended by Illumina (Table 3).

The V4 region of the 18S rDNA of eukaryotes was amplified using the following nested PCR assays. The 1st PCR (UNonMet-PCR) reaction was performed to decrease the amplification of host DNA fragments in the blood samples using the primers 18S-EUK581-F and 18S-EUK1134-R [64]. The resulting PCR products were used as DNA templates for a 2nd PCR using the primers Illumina_E572F and Illumina_E1009R [65]. Each PCR reaction contained 12.5 μL of 2× KAPA HiFi HotStart ReadyMix (Kapa BioSystems, Wilmington, MA, USA); 5 μL of each primer and 2.5 μL of the genomic DNA samples or 1st PCR products. The UNonMet-PCR included 15 cycles followed by the 2nd PCR using 30 cycles as shown in Table 3. PCR results were visualized by electrophoresis on 1.5% agarose gel stained with Gel-RedTM (Biotium, Hayward, CA, USA). Illumina sequencing libraries were prepared by purifying the amplicons using AMPure XP (Beckman Coulter Life Sciences, IN, USA) and sequencing adapters and index sequences were added using the Nextera XT Index Kit (Illumina). The sequencing run was conducted with a MiSeq Reagent Kit v3 (600 cycles) on an Illumina MiSeq device according to the manufacturer’s instructions.

### 5.3. Illumina Data Processing

The resulting fastq files were analyzed in QIIME2 (version 2019.10.0) [66]. The forward and reverse reads were merged into one sequence and the obtained sequences were quality checked and filtered. A feature table was established using the Divisive Amplicon Denoising Algorithm 2 (DADA2) pipeline [67]. We used silva-132-99-nb classifier to assign taxonomy to each ASV. Differential abundances of the detected taxonomic groups were visualized using the taxa_heatmap function in the qiime2R package in R (version 2.13.0). Contaminants were filtered in R (version 3.4.1) by using the prevalence-based method in Decontam R package by setting the threshold = 0.5 and indicating the negative control sample as “negative”. [68]. Indicated contaminants were filtered out from all samples in QIIME2. Raw sequence data have been deposited in DNA Data Bank of Japan (DDBJ) sequence read archive with an accession number of DRA011269. 

### 5.4. Conventional PCR and Sanger Sequencing

A total of 100 camel blood samples were tested by conventional PCRs targeting 18S rDNA (RLB-PCR), ITS1 region and 16S rDNA (EHR-PCR) of apicomplexan protozoa, trypanosomes and Anaplasmataceae, respectively. *Trypanosoma*-positive samples were further examined for *Tr. evansi* by ILO-PCR using the primers ILO7957 and ILO8091 [16] and by EVAB-PCR using the primers EVAB1 and EVAB2 [15]. *Anaplasma*-positive samples were further characterized by amplifying a 650 bp and 1100 bp segments of *gltA* and *groEL* genes, respectively, as previously described [69].

PCR reactions consisted of a 25 μL mixture of 12.5 μL of 2× Gflex PCR Buffer, 0.5 μL of Tks Gflex DNA Polymerase (1.25 units/μL) (TaKaRa Bio Inc., Shiga, Japan), 0.5 μL of each primer, 1.0 μL of template DNA and molecular grade water (Table 3). The PCR System 9700 (Applied Biosystems, Foster City, CA, USA) was used to run an initial denaturation at 94 °C for 1 min, followed by 40 cycles of denaturation at 94 °C for 10 s, annealing (Table 3), extension at 68 °C for 15 s and final extension at 68 °C for 5 min. The primer sequences used in the present study are listed in Table 3. PCR products were analyzed by electrophoresis in 1% agarose gel stained with Gel-Red (Biotium, Hayward, CA, USA) and visualized under an ultraviolet (UV) light.

The positive PCR products that were obtained from RLB-PCR, EHR-PCR, *gltA*-PCR and *groEL*-PCR were purified by ExoSAP-IT PCR Product Cleanup Reagent (Applied Biosystems, Foster City, CA, USA) and sequenced using the BigDye Terminator v3.1 Cycle Sequencing Kit (Applied Biosystems) and an ABI Prism 3130xl genetic analyzer (Applied Biosystems).

### 5.5. Phylogenetic Analysis

The obtained 18S rDNA features belonging to genus *Theileria* were exported from QIIME2 (version 2019.10.0) [66] to Geneious v10.2.6 (Biomatters Ltd., Auckland, New Zealand). In addition, the obtained sequences from Sanger sequencing were also assembled in Geneious v10.2.6 (Biomatters Ltd., Auckland, New Zealand) and the primer regions were removed. We compared the obtained sequences to those in published databases using the nucleotide basic local alignment search tool (BLASTn, https://blast.ncbi.nlm.nih.gov/Blast.cgi?PAGE_TYPE=BlastSearch) (accessed on 12 December 2020). The sequences were aligned using MAFFT v7.450 software [70] and the best fit model for the analysis was determined using MEGA X software [71] and phylogenetic trees were constructed using the maximum likelihood method in PHYML v3.3 software [72].

The sequences obtained in this study were submitted to the DDBJ under the accession numbers: LC592651-LC592666 for 18S rDNA, LC592622-LC592650 for 16S rDNA, LC592667-LC592677 for *gltA* and LC592678-LC592688 for *groEL*.

## Figures and Tables

**Figure 1 pathogens-10-00351-f001:**
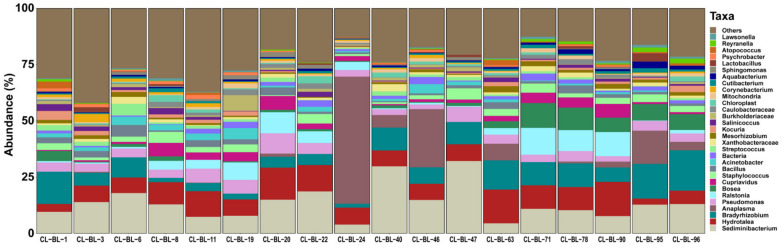
Relative abundance of bacterial genera detected in 18 camel blood samples. Each bar represents the bacterial taxa detected in one camel sample. The sample ID is provided on the bottom of each bar.

**Figure 2 pathogens-10-00351-f002:**
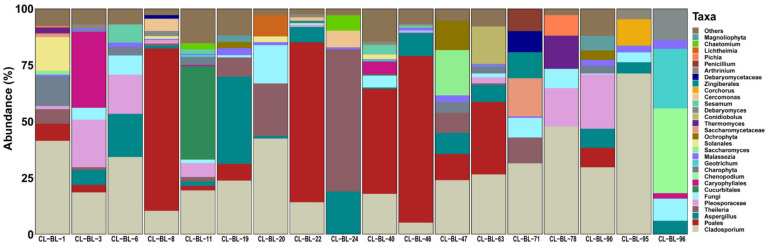
Relative abundance of eukaryotic genera detected in 18 camel blood samples. Each bar represents the eukaryotic taxa detected in one camel sample. The sample ID is provided on the bottom of each bar.

**Figure 3 pathogens-10-00351-f003:**
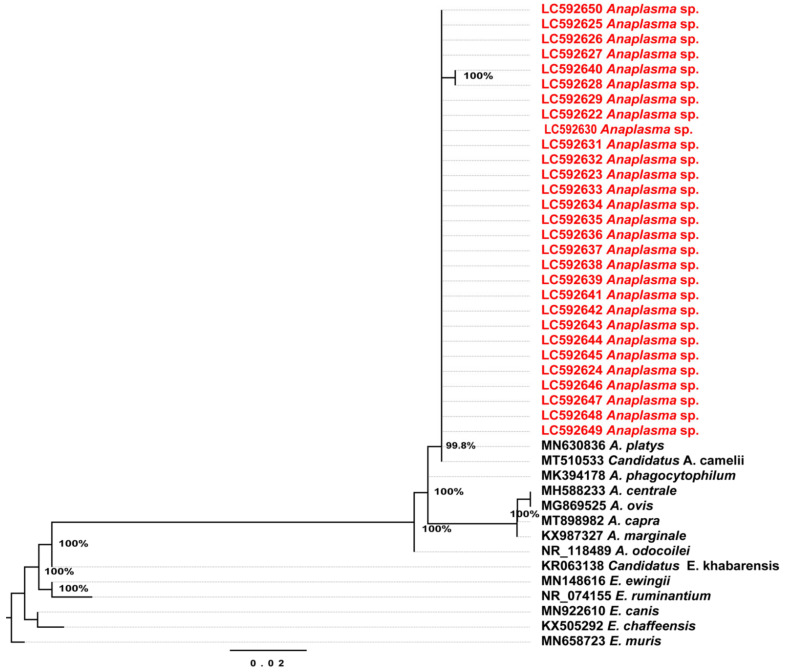
Phylogenetic tree based on the 16S rDNA sequences of *Anaplasma* and *Ehrlichia*. The tree was constructed by the maximum likelihood (ML) method based on the HKY85 model (Hasegawa–Kishino–Yano, 85) and site heterogeneity model (Invariant Sites) using the PhyML program. Numbers at the nodes are bootstrap values supported from 1000 replications. The scale bar represents 0.02 nucleotide substitutions per nucleotide site. Red font labels indicate GenBank accession numbers of 16S rDNA of *Candidatus* Anaplasma camelii sequences obtained in this study. The tree was rooted to *Ehrlichia muris* (MN658723).

**Figure 4 pathogens-10-00351-f004:**
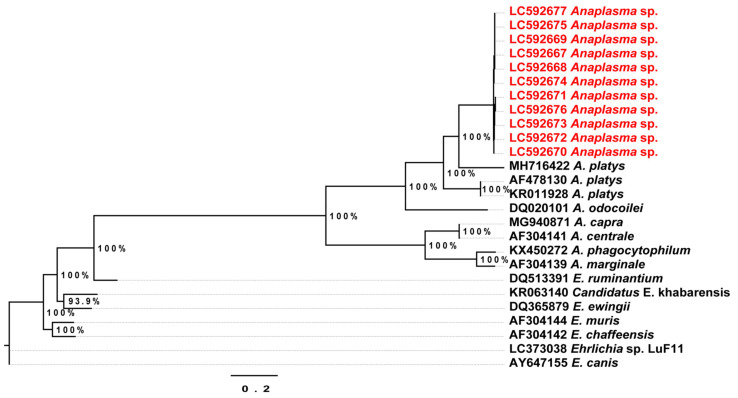
Phylogenetic tree based on the *gltA* gene sequences of *Anaplasma* and *Ehrlichia*. The tree was constructed by the maximum likelihood method based on the GTR (general time reversible) model and site heterogeneity model (Gamma + Invariant Sites) using PhyML program. Numbers at the nodes are bootstrap values supported from 1000 replications. The scale bar represents 0.2 nucleotide substitutions per nucleotide site. Red font labels indicate GenBank accession numbers of the *gltA* gene of *Candidatus* Anaplasma camelii sequences from this study. The tree was rooted to *Ehrlichia canis* (AY647155).

**Figure 5 pathogens-10-00351-f005:**
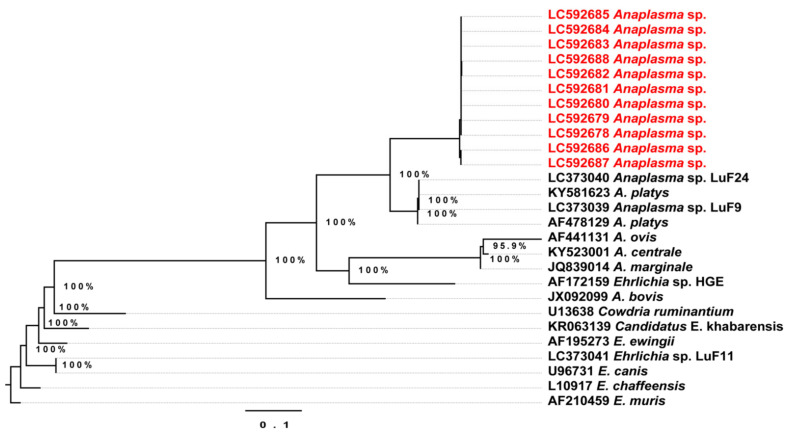
Phylogenetic tree based on the *groEL* gene sequences of *Anaplasma* and *Ehrlichia*. The tree was constructed by the maximum likelihood method based on the GTR (general time reversible) model and site heterogeneity model (Gamma + Invariant Sites) using PhyML program. Numbers at the nodes are bootstrap values supported from 1000 replications. The scale bar represents 0.1 nucleotide substitutions per nucleotide site. Red font labels indicate GenBank accession numbers of the *gltA* gene of *Candidatus* Anaplasma camelii obtained in this study. The phylogenetic tree was rooted to *Ehrlichia muris* (AF210459).

**Figure 6 pathogens-10-00351-f006:**
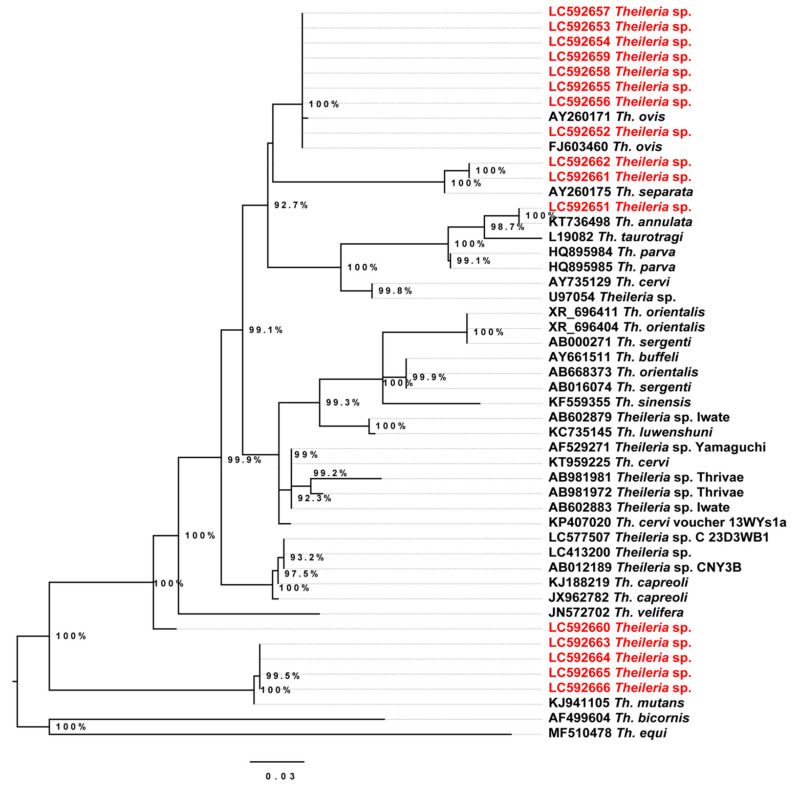
Phylogenetic tree based on the 18S rDNA sequences of *Theileria*. The tree was constructed by the maximum likelihood method based on the GTR (general time reversible) model and site heterogeneity model (Gamma + Invariant Sites) using PhyML program. Numbers at the nodes are bootstrap values supported from 1000 replications. The scale bar represents 0.03 nucleotide substitutions per nucleotide site. Red font labels indicate GenBank accession numbers of the 18S rDNA sequences of *Theileria* spp. obtained in this study. The tree was rooted to *Theileria equi* (MF510478).

**Table 1 pathogens-10-00351-t001:** A summary of the number of reads originating from the tick-borne pathogens detected in this study.

Sample ID	*Candidatus* Anaplasma Camelii	*Theileria annulata*	*Theileria ovis*	*Theileria separata*	*Theileria mutans*-Like *	*Theileria* sp.
CL-BL-1	28	0	488 (LC592658) **	0	241 (LC592666)	65 (LC592660)
CL-BL-3	0	0	171 (LC592657)	0	0	0
CL-BL-6	0	0	0	0	0	0
CL-BL-8	324	0	175 (LC592656)	0	0	0
CL-BL-11	0	0	120 (LC592659)	0	0	0
CL-BL-19	13,101	314 (LC592651)	434 (LC592655)	0	0	0
CL-BL-20	128	0	1252 (LC592654)	0	572 (LC592665)	0
CL-BL-22	931	0	0	0	0	0
CL-BL-24	1914	0	2945 (LC592653)	626 (LC592662)	1360 (LC592664)	0
CL-BL-40	0	0	0	0	0	0
CL-BL-46	909	0	0	0	0	0
CL-BL-47	0	0	0	558 (LC592661)	0	0
CL-BL-63	0	0	0	0	77 (LC592663)	0
CL-BL-71	88	0	513 (LC592652)	0	0	0
CL-BL-78	0	0	0	0	0	0
CL-BL-90	461	0	0	0	0	0
CL-BL-95	2738	0	0	0	0	0
CL-BL-96	589	0	0	0	0	0

* *Theileria mutans*-like was identified as *Theileria* sp. strain MSD by silva-132-99-nb classifier in QIIME2. ** GenBank accession numbers are shown in brackets.

**Table 2 pathogens-10-00351-t002:** A comparison of results from next-generation sequencing (NGS)-based and conventional polymerase chain reaction (PCR)-based methods in detecting *Anaplasma* spp., *Theileria* spp. and *Trypanosoma evansi* in camel blood samples.

Sample ID	*Anaplasma* spp.				*Theileria* spp.		*Trypanosoma evansi*	
	NGS (V3-V4-PCR)	EHR-PCR	*gltA*-PCR	*groEL*-PCR	NGS (UNonMet-PCR)	RLB-PCR	NGS (UNonMet-PCR)	ILO-PCR
CL-BL-1	N	N	N	N	P	N	N	N
CL-BL-3	P	N	N	N	P	N	N	N
CL-BL-6	N	N	N	N	N	N	N	N
CL-BL-8	N	N	N	N	P	N	N	P
CL-BL-11	P	N	N	N	P	N	N	N
CL-BL-19	N	N	N	N	P	N	N	P
CL-BL-20	P	N	N	N	P	N	N	N
CL-BL-22	N	N	N	N	N	N	N	N
CL-BL-24	P	N	N	N	P	N	N	N
CL-BL-40	P	P	P	P	N	N	N	P
CL-BL-46	P	P	P	P	N	N	N	P
CL-BL-47	N	N	N	N	P	N	N	P
CL-BL-63	P	N	N	N	P	N	N	N
CL-BL-71	N	N	N	N	P	N	N	N
CL-BL-78	P	N	N	N	N	N	N	N
CL-BL-90	P	P	P	P	N	N	N	P
CL-BL-95	P	P	P	P	N	N	N	N
CL-BL-96	P	P	P	P	N	N	N	N

V3-V4-PCR, PCR targeting the 16S rRNA gene of prokaryotes; EHR-PCR, PCR targeting the 16S rRNA gene of Anaplasmataceae; *gltA*-PCR, PCR targeting the *gltA* gene of Anaplasmataceae; *groEL*-PCR, PCR targeting the *groEL* gene of Anaplasmataceae; UNonMet-PCR, PCR targeting the 18S rRNA gene of nonmetazoans; RLB-PCR, PCR targeting the 18S rRNA gene of apicomplexan protozoa; ILO-PCR, PCR targeting the RoTat1.2 VSG region; N, negative; P, positive.

**Table 3 pathogens-10-00351-t003:** List of PCR primers used in this study.

PCR Name	Primer Name	Primer Sequence (5′–3′)	Annealing Temp/Extension Time	Reference
V3-V4-PCR	Illumina_16S_341F	TCGTCGGCAGCGTCAGATGTGTATAAGAGACAGCCTACGGGNGGCWGCAG	55 °C/30 s	[54]
	Illumina_16S_805R	GTCTCGTGGGCTCGGAGATGTGTATAAGAGACAGGACTACHVGGGTATCTAATCC		
UNonMet-PCR	18S-EUK581-F	GTGCCAGCAGCCGCG	62 °C/30 s	[22]
	18S-EUK1134-R	TTTAAGTTTCAGCCTTGCG		
V4-PCR	Illumina_E572F	TCGTCGGCAGCGTCAGATGTGTATAAGAGACAGCYGCGGTAATTCCAGCTC	55 °C/30 s	[23]
	Illumina_E1009R	GTCTCGTGGGCTCGGAGATGTGTATAAGAGACAGAYGGTATCTRATCRTCTTYG		
RLB-PCR	RLB-F2	GACACAGGGAGGTAGTGACAAG	54 °C/15 s	[55]
	RLB-R2	CTAAGAATTTCACCTCTGACAGT		
ITS1-PCR	ITS1-CF	CCGGAAGTTCACCGATATTG	52 °C/15 s	[56]
	ITS1-BR	TTGCTGCGTTCTTCAACGAA		
EHR-PCR	EHR16SD	GGTACCYACAGAAGAAGTCC	61 °C/15 s	[57]
	EHR16SR	TAGCACTCATCGTTTACAGC		
*gltA*-PCR (1st)	F4b	CCGGGTTTTATGTCTACTGC	55 °C/15 s	[58]
	R1b	CGATGACCAAAACCCAT		
*gltA*-PCR (2nd)	EHR-CS136F	TTYATGTCYACTGCTGCKTG	50 °C/15 s	[58]
	EHR-CS778R	GCNCCMCCATGMGCTGG		
*groEL*-PCR (1st)	HS1-F	CGYCAGTGGGCTGGTAATGAA	54 °C/15 s	[59,60]
	HS6-R	CCWCCWGGTACWACACCTTC		
*groEL*-PCR (2nd)	HS3-F	ATAGTYATGAAGGAGAGTGAT	50 °C/15 s	[60,61]
	HSV-R	TCAACAGCAGCTCTAGTWG		
ILO-PCR	ILO7957	GCCACCACGGCGAAAGAC	52 °C/15 s	[62]
	ILO8091	TAATCAGTGTGGTGTGC		
EVAB-PCR	EVAB1	CACAGTCCGAGAGATAGAG	60 °C/15 s	[63]
	EVAB2	CTGTACTCTACATCTACCTC		

## Data Availability

Raw sequence data have been deposited in the DNA Data Bank of Japan (DDBJ) Sequence Read Archive with an accession number of DRA011269.

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
