# Peer review of "Exploring Prokaryotic and Eukaryotic Microbiomes Helps in Detecting Tick-Borne Infectious Agents in the Blood of Camels"

_pathogens, 2021, doi:10.3390/pathogens10030351_

Round 1

Reviewer 1 Report

In this study, Wessam et al. combined a conventional PCR based approach and metagenomics to detect procaryotic and protozool sequences in blood samples of camels collected during import quarantine.

This study is technically sound, however I have some remarks regarding data presentation, study design and conclusions:

  1. The authors collected 100 blood samples and tested them by conventional PCR assays, of those samples 18 were selected for 16S and 18S metagenomics using HTS. It is not clear based on which criteria these 18 samples were selected. The order of presentation is first metagenomics on the 18 samples and then PCR on all 100 samples. I would prefer to change this order and to explain to the reader why these samples were collected for NGS based metagenomics.
  2. I think that the authors overinterpret the detection of pro- and eukaryotic sequences in blood by metagenomics. They conclude that there is a whole microbiome circulating in the blood. But this is based only on the classification  of NGS reads. The authors should be more cautious in interpreting these data, without confirmimg these findings with other techniques, such as cultures or more specific PC assays. They should also discuss possilbe introduction of these seqeunces  during sampling, sample storage and sample processing.
  3. lines 61-78 ff: it is intersting to read which strategoes have been applied to reduce amplification of host 18S rDNA in metagenomic studies, however at this stage of the introduction it is not so clear how this relates to the present study.  Please explain.
  4. Figures 1 and 2: please explain the labeling of the samples and the meaning of CL-BL-Neg
  5. Figure 3 and 4:are the red labelled sequence numbers Genebank Accession numbers? PLease eplain. Could you please link the sequences in Fig 3 to the animals and Theileria sp. in Table 1.

Author Response

Reviewer 1:

In this study, Wessam et al. combined a conventional PCR based approach and metagenomics to detect procaryotic and protozool sequences in blood samples of camels collected during import quarantine.

This study is technically sound, however I have some remarks regarding data presentation, study design and conclusions:

Authors: Thank you so much for reviewing our paper and providing us with the valuable suggestions and corrections. We responded to the reviewer’s suggestions and corrections. We hope our manuscript will be acceptable after making the corrections and changes. We are attaching our responses as follow:

  1. The authors collected 100 blood samples and tested them by conventional PCR assays, of those samples 18 were selected for 16S and 18S metagenomics using HTS. It is not clear based on which criteria these 18 samples were selected. The order of presentation is first metagenomics on the 18 samples and then PCR on all 100 samples. I would prefer to change this order and to explain to the reader why these samples were collected for NGS based metagenomics.

Authors: Thank you so much for this suggestion. As we mentioned in Introduction, NGS-based approach allows simultaneous detection of known pathogens. Theoretically, it can also detect novel potential pathogens. However, NGS-based approach only amplifies a short region of genetic markers (16S rDNA and 18S rDNA), which is not always sufficient to characterize pathogens. This means that subsequent PCR based assays and Sanger sequencing are required. In this paper we aimed to detect multiple pathogens including previously unrecognized ones and characterized by conventional PCR assays.

Due to the limitation of the capacity for NSG analysis, we could only test 18 samples which were randomly selected from 100 camel samples. Since all animals were adult males imported from Sudan without any clinical sings, we could not make any stratified sample selection.

We included the following sentences to clarify the above-mentioned strategy and method.

(Line 201): “Our strategy was to first detect potential pathogens by NGS-based approach in the randomly selected camel samples and to confirm the results by testing all camel samples using target specific conventional PCR assays.”

(Line 323): “Since all camels showed no clinical signs, we randomly selected these samples.”

  1. I think that the authors overinterpret the detection of pro- and eukaryotic sequences in blood by metagenomics. They conclude that there is a whole microbiome circulating in the blood. But this is based only on the classification  of NGS reads. The authors should be more cautious in interpreting these data, without confirmimg these findings with other techniques, such as cultures or more specific PC assays. They should also discuss possilbe introduction of these seqeunces  during sampling, sample storage and sample processing.

Authors: We understand the reviewer’s concern. We resolved this issue by adding the following to the discussion section (Line 218-221): “However, there is still a possibility that environmental bacteria might have been introduced during sampling and laboratory procedures. Therefore, more experimental investigations are required to confirm the presence of these bacteria in the camel blood by employing culture-based and target specific PCR-based methods.”.

  1. lines 61-78 ff: it is intersting to read which strategoes have been applied to reduce amplification of host 18S rDNA in metagenomic studies, however at this stage of the introduction it is not so clear how this relates to the present study.  Please explain.

Authors: We moved this paragraph to the discussion section (Line 288-300).

  1. Figures 1 and 2: please explain the labeling of the samples and the meaning of CL-BL-Neg

Authors: We indicated the sample id including the CL-BL_Neg by adding the following to the legends:

Fig 1 (Line 115-116): “Each bar represents the bacterial taxa detected in one camel sample. The sample ID is provided on the bottom of each bar. CL-BL-Neg stands for negative control.”

Fig 2 (Line 118-119): “Each bar represents the eukaryotic taxa detected in one camel sample. The sample ID is provided on the bottom of each bar. CL-BL-Neg stands for negative control..”

  1. Figure 3 and 4:are the red labelled sequence numbers Genebank Accession numbers? PLease eplain. Could you please link the sequences in Fig 3 to the animals and Theileria sp. in Table 1.

Authors: We referred to the red labelled sequence as GenBank accession numbers. We added

(Line 128 ): “Red font labels indicate GenBank accession numbers of the 18S rDNA sequences of Theielria spp. obtained in this study”

 (Line 143 ): “Red font labels indicate GenBank accession numbers of 16S rDNA of Candidatus A. camelii sequences obtained in this study.”

We also linked the sequences in Fig 3 to the animals and Theileria spp. in table. We added the GenBank accession numbers in Table 1 (Line 120).

Reviewer 2 Report

My main comment is that the authors submitted the manuscript to the Ticks section. While the manuscript does not focus on the role of ticks in the circulation of pathogens. The authors only mention that, referring to the literature, certain species of ticks were collected from camels. They also list the pathogens they identified, suggesting that the animals may have been infected by ticks. Despite this, I find the article interesting, the authors complied with the publisher's editorial requirements. I only suggest minor revisions.

L37 please add information on how many animals have been tested

Section 5.1 is not needed because the authors have included the consent information at the end of the manuscript as per the publisher's requirements.

Author Response

Reviewer 2:

  • My main comment is that the authors submitted the manuscript to the Ticks section. While the manuscript does not focus on the role of ticks in the circulation of pathogens. The authors only mention that, referring to the literature, certain species of ticks were collected from camels. They also list the pathogens they identified, suggesting that the animals may have been infected by ticks. Despite this, I find the article interesting, the authors complied with the publisher's editorial requirements. I only suggest minor revisions.

Authors: Thank you for reviewing our article, we will consider all your valuable comments.

We selected the Ticks section because most of the detected organisms are known to be tick-borne. Although we have not examined the attached ticks, we believe that information provided in our study will guide many future studies on tick-borne diseases in this region.

  • L37 please add information on how many animals have been tested

Authors: Thank you for your precise addition, I added the number of samples in L37

  • Section 5.1 is not needed because the authors have included the consent information at the end of the manuscript as per the publisher's requirements.

Authors: Thanks for the valuable comment, we removed section 5.1 from the manuscript and corrected the subheading numbering.